# Quantification of Enhydrin and Uvedalin in the Ethanolic Extract of *Smallanthus sonchifolius* Leaves Validated by the HPLC Method

**DOI:** 10.3390/molecules28041913

**Published:** 2023-02-17

**Authors:** Hady Anshory Tamhid, Triana Hertiani, Yosi Bayu Murti, Retno Murwanti

**Affiliations:** 1Faculty of Pharmacy, Universitas Gadjah Mada (UGM), Yogyakarta 55281, Indonesia; 2Department of Pharmacy, Universitas Islam Indonesia (UII), Yogyakarta 55581, Indonesia

**Keywords:** *Smallanthus sonchifolius*, yacon, enhydrin, uvedalin, HPLC, validation

## Abstract

Yacon leaf (*Smallanthus sonchifolius*, Asteraceae) ethanolic extracts are widely used in herbal medicine preparation for diabetes. They contain two sesquiterpene lactones (enhydrin (**1**) and uvedalin (**2**)) as major bioactive compounds. To provide a suitable method of analysis for the extract’s quality control, we developed and validated a simultaneous HPLC-UV method using the compounds as markers. Compounds **1** and **2** were isolated using a freeze crystallization technique followed by a preparative HPLC. Spectrometry data for **1** and **2** were determined and compared to the literature. Chromatographic separation was carried out for 30 min with a mobile phase that used 60% water and 40% acetonitrile and a C18 column (250 × 4.6 mm, 5 µm) as the stationary phase. The flow was set to 1 mL min^−1^ and detection was conducted at 210 nm. The validation method was conducted according to the ICH guidelines, which included linearity, precision, accuracy, LOD, and LOQ. The calibration curve of both compounds was linear (R ^2^ > 0.9999), with the limit of detection and quantification as follows, respectively, 0.52 and 1.57 µg/mL for **1**, and 0.144 and 0.436 µg/mL for **2**. The percentages of recovery and repeatability (%RSD) were, 101.46 and 0.30% for **1**, and 97.68 and 0.08% for **2**, respectively. The **1** and **2** were 1.67 and 0.88% in the Ykal extract, and 1.26 and 0.56% in the Ycin extract, respectively. The method was found to be linear, precise, accurate, and suitable to be applied for control quality analyses of yacon leaf extract.

## 1. Introduction

Yacon (*Smallanthus sonchifolius*) leaves have been used as a traditional medicine to treat diabetes [1]. Scientifically, yacon leaf extract was reported to reduce blood sugar levels in rats induced with streptozotocin (STZ) and increase insulin levels in the blood [2]. In addition, yacon leaf extracts is known to inhibit the action of the enzyme Diphenyl peptidase 4 (DPP-IV) [3] and are thought to increase cell sensitivity to insulin [4].

Yacon is native to the Andean regions of South America. The name of yacon is derived from “yakku”, a Quechua language that means tasteless/watery [5]. Yacon is classified into sunflower plants and is included in the Compositae or Asteraceae [6].

The flowers are very similar to sunflowers, except that they are smaller in size, with a diameter of approximately 4–5 cm. The difference is very far compared to sunflowers, with flower diameters reaching 30 cm. The flower structure and color are the same. The leaves of this plant are heart-shaped, with a length of up to 50 cm from the tip of the stalk and a width of up to 40 cm. The stem of the yacon plant is light green to purple, with a diameter of 2–5 cm. This plant has tubers with an enormous size that reaches the size of an adult’s arm. The average weight of one tuber reaches 1–2 kg and 5–10 tubers can be obtained in one plant. The height of this plant is about 1–2 m and can reach 3 m. It lives well in cold temperatures at an altitude of 500 m above sea level. The age of the plant until the roots are formed optimally is 6–7 months [7].

According to Honore (2015), there are two main types of compounds found in yacon leaves: phenolic acid and melampolide [8]. Sesquiterpene lactones are an important group of natural products obtained from Asteraceae, especially the *Smallanthus sonchifolius* species. Sesquiterpene lactones were accumulated mainly in the leaves of yacon [9]. Enhydrin and uvedalin are the most dominant types of sesquiterpene lactones found in yacon leaves [10,11]. Enhydrin and uvedalin levels in yacon leaves are 0.74 mg/g and 0.21 mg/g of fresh material, respectively [12]. The enhydrin content of dry leaves can reached 0.97% [13]. Enhydrin compounds are active compounds from yacon leaves and can be used as marker compounds for the quality control of yacon plants as anti-diabetic agents [14]. Enhydrin is known to have activity in lowering blood sugar levels in vivo [15], whereas uvedalin is known to have antibacterial activity against *Bacillus anthracis* and MRSA [16]. Both types of sesquiterpene lactone compounds are essential in the activity of yacon leaves being anti-diabetic and antibacterial. The ratio content of enhydrin and uvedalin in the extract of yacon leaves is known to affect its action as an antibacterial [17]. Of the ten sesquiterpene lactone compounds isolated from yacon leaves, only enhydrin and uvedalin show inhibitory activity against the NF-kB enzyme, which regulates the immune system and inflammation [18]. Enhydrin is a chemotype of yacon leaves, while uvedalin is its subtype [19]. These two compounds are discriminant substances that can be used as the basis for quality control of yacon leaf extract as a traditional medicinal ingredient. So, it is crucial to see the content of enhydrin and uvedalin in yacon leaf extract as active compounds. However, there has never been a simultaneous determination of the enhydrin and uvedalin quantification using a validated method in yacon leaf extract as part of quality control.

HPLC is a compound analysis method with high accuracy and precision in determining the levels of compounds in plant extracts. Quantifying single enhydrin compounds using the HPLC method has been reported [13], but the sample used is leaf rinse extract. However, commercially, especially in Indonesia, yacon leaf extracts used as a traditional medicine are extracted from dried yacon leaf powder with a solvent in accordance with regulatory recommendations [20]. In the current study, we developed a simple method, validated by to determine enhydrin and uvedalin levels simultaneously. This method can be suitable to control the quality of yacon leaf extracts. It is handy for determining the quality of yacon leaf extracts as a raw material for traditional medicine, especially for its use as an anti-diabetic herbal medicine.

## 2. Results and Discussion

### 2.1. Characterization of Compound Isolates

Sesquiterpene lactones, enhydrin, and uvedalin are active compounds in yacon leaves that have various activities. These compounds are collected in glandular trichomes on the leaf’s surface [21]. Rinse extraction is a suitable method for extracting compounds on a leaf’s surface. The rinsate obtained from this extraction containsed more enhydrin and uvedalin compounds. Furthermore, freeze crystallization at −20 °C was performed to isolate the two compounds. Separation of the two compounds was carried out by preparative HPLC with solvent acetonitrile: water gradient on C18 column (150 mm × 7.8 mm, 5 µm). Chromatograms of the two compounds are shown in Figure 1A,B. Compounds **1** and **2** were characterized using LC-MS and NMR spectroscopy, and the results were compared with reference standards. The characteristics of ^1^H and ^13^C NMR spectra of these compounds show identical characters to the standard reference.

Compound **1** is in the form of colorless crystals with the molecular formula C_23_H_28_O_10_, and a molecular weight of m/z 464.1677 (calc. 464.1733) based on LC-MS results (Appendix A). Meanwhile, compound **2** is amorphous with a molecular formula of C_23_H_28_O_9_, and a molecular weight of 448.1728 (calc. 448.1787) based on the LC-MS results (Appendix A). The ^1^H and ^13^C NMR data of compounds **1** and **2** have characteristics as sesquiterpene lactones with the same comparison as the literature [22,23]. Both compounds have the same structural framework, namely sesquiterpene lactones. The characteristics of compounds **1** and **2** based on ^1^H and ^13^C NMR data are very identical to the results shown in the literature [22,23]. The difference between these two compounds lies in the epoxy group, where compound **1** has two epoxy groups, whereas compound **2** has only a single epoxy group (Figure 2). The difference in the epoxy groups can be seen in the carbon 13 signal shown in C-4 and C-5 (Table 1).

There are two typical doublets in the ^1^H NMR spectrum at δ 5.87 and 6.34 of compound **1** and δ 5.73 and 6.26 of compound **2,** which are typical for the exocyclic methylene protons of the lactone group. The ^13^C NMR data indicated that four carbonyl carbon signals at δ 168.0 (C-12), 165.5 (C-14), 168.4 (C-1′), and 170.4 (C-Ac), and four olefinic carbon signals at δ 149.4 (C-1), 130.1 (C-10), 133.3 (C-11), and 122.9 (C-13) of compound **1** were present. Meanwhile, For compound **2**, four carbonyl carbon signals at δ 168.5 (C-12), 165.9 (C-14), 169.1 (C-1′), and 170.2 (C-Ac), and six olefinic carbon signals at δ 148.38 (C-1), 130.64 (C-4), 126.14 (C-5), 134.48 (C-10), 138.68 (C-11), and 121.55 (C-13) were present (Appendix A). All the proton and carbon signals of compounds **1** and **2** were identical to the references [22,23].

Enhydrin (**1**): ^1^H NMR (500 MHz, CDCl3): 7.15 (1H, dd), 6.70 (1H, d), 6.34(1H, d), 5.87(1H, d), 5.84(1H, d), 4.27(1H, t), 3.83 (3H, s), 3.00 (3H, m), 2.68 (1H, d), 2.45 (1H, m), 2.35 (1H, m), 2.05 (3H, s), 1.71 (3H, s), 1.45 (3H, s), 1.22 (1H, d), and 1.19 (3H, m).

Uvedalin (**2**): ^1^H NMR (500 MHz, CDCl3): 7.05 (1H, dd), 6.66 (1H, dd), 6.26 (1H, d), 5.73 (1H, d), 5.41 (1H, d), 5.11 (1H, t), 4.96 (1H, d), 3.80 (3H, s), 3.02 (1H, dd), 2.79 (1H, m), 2.67 (1H, m), 2.42 (3H, m), 2.00 (3H, s), 2.00 (3H, s), 1.47 (3H, s), and 1.19 (3H, d).

### 2.2. HPLC Method Validation

#### 2.2.1. Linearity, LOD, and LOQ

Linearity can be observed through a calibration curve showing the relationship between the response and the analyte concentration in several series of standard solutions. This calibration curve calculates the linear regression in the formula
*y* = *bx* + *a*,(1)
where *x* is the concentration, *y* is the response, *a* is the true y-intercept, and *b* is the true slope. This regression aims to determine the best estimate for the slope and *y*-intercept so that it will minimize the residual error, which is the difference between the experimental value and the value predicted by the linear regression function.

Concentrations testing the linearity of compounds **1** and **2** used five variations: 5, 10, 25, 50, and 100 µg/mL. The slope, *y*-intercept, and correlation coefficient (R^2^) are shown in Table 2. The calibration curve obtained with the concentration variations was linear. The linear regression of compounds **1** and **2** are as follows: *y* = 40053*x* + 6383.6 (R^2^ = 0.9999) (2)
*y* = 41291.33*x* + 4172.92 (R^2^ = 0.9999) (3)

As a parameter for the existence of a linear relationship, the correlation coefficient (R^2^) in linear regression analysis is used. An ideal linear relationship is achieved if the R^2^ value is above 0.9990. The correlation coefficient (R^2^) of both is greater than 0.9990. It indicates that the degree of correlation is high and this method has good linearity.

The limit of detection (LOD) is the lowest analyte concentration in a sample that can be detected but not measured to its true value. The limit of quantification (LOQ) is the lowest concentration of analyte in a sample that can be quantitatively measured with good accuracy and precision. LOQ is a quantitative testing parameter for low analyte concentrations in complex matrices and is used to determine the presence of impurities or product degradation. LOD and LOQ were calculated from equations that consider the parameters of the analytical curve, using the standard deviation of the response and the slope of the analytical curve.

The LOD and LOQ values can be used to determine the sensitivity of a test method. The LOD and LOQ values obtained were 0.52 and 1.57 µg/mL for 1, and 0.144 and 0.436 µg/mL for compound **2**, respectively (Table 2). These results indicate that the method provided adequate sensitivity.

#### 2.2.2. Precision and Accuracy

Precision is the closeness of the results obtained from a series of repeated measurements of the same measure. The precision determination type in this research is repeatability, which is the preciseness determined by one analyst on the same day with the same equipment and laboratory.

The intra-day precision for determining compounds **1** and **2** is given in Table 3. The %RSD values of intra-day precision (repeatability) were 0.02–0.53% (0.3%) and 0.02–0.14% (0.08%) for compounds **1** and **2**, respectively. A method is precise if the %RSD value is less than 2%. The precision of the validation results shows a small value, reflecting the high precision of this method.

Accuracy is a measure that indicates the closeness of the analytical results to the actual standard concentration. Accuracy is expressed as a percent recovery of the addition standard. Accuracy is the exactness of the analytical method or the closeness between the measured value and the accepted value, either the convention value, the actual value, or the reference value. Accuracy is calculated as the recovery of standard in a measurement by spiking a sample. In this work, the standards were added at concentrations of 5, 10, and 25 µg/mL, as shown in Table 4.

The accuracy value is expressed as a recovery percentage (%R). Accuracy measurement is accomplished by adding a standard with a particular concentration to a sample (spike). The results of the determination of recovery are shown in Table 4. The average recovery yields (%R) obtained for compounds **1** and **2** were 101.46 and 97.68%, respectively. This result shows good accuracy because it is included in terms of acceptance of percent recovery in the range of 95–105% [24].

### 2.3. HPLC Method Application for the Quantification of Compounds 1 and 2

Quantifying enhydrin and uvedalin using the HPLC method is suitable for application as a quality control for yacon leaf extracts. The chromatogram obtained by the HPLC system showed good separation. The combination of enhydrin and uvedalin also showed good separation by HPLC (Figure 1C). In addition, the method that has been subjected to the validation process was found to be linear, precise, and accurate.

This method was used to calculate the levels of enhydrin and uvedalin compounds in the ethanolic extracts of yacon leaves obtained from two regions. The enhydrin and uvedalin compounds are the chemotype and subtype compounds used in determining the quality of yacon leaf extract as a traditional medicinal ingredient [19]. They are mostly stored in the trachoma glandular on the leaf’s surface [21]. However, the content of these two compounds can deviate depending on the growth location. Comparing the contents of these two compounds in varieties and various locations of growth in Japan shows that the levels of enhydrin and uvedalin in the extract range between 1–7 and 0.5–2%, respectively [12]. In the current study, the levels of enhydrin and uvedalin obtained from two different areas on the island of Java, Indonesia, were also different. The content of the enhydrin compounds obtained in the Ykal and Ycin samples were 1.67 and 1.26%, respectively, while uvedalin was 0.88 and 0.59% (Table 5). The levels of the two compounds were relevant to the above study. Even though the factors that cause the disparity cannot be explained with certainty, the difference in the highlands where they grow influences the content of the two compounds. The profile of the compound/metabolite content of the yacon plants taken from the two regions in Indonesia also showed different results [25].

Yacon leaf metabolites collected starting from leaves that were 15 days old after sprouting and then harvested every 15 days to 6 months were analyzed. The results showed that, of all the STL compounds identified, only enhydrin and uvedalin were always present in all of these samples. It suggests that the possibility of biosynthesis of enhydrin and uvedalin starts from the beginning of leaf growth and is stable until the plant period ends [26]. In addition, the presence of these compounds in yacon leaves plays a role in anti-diabetic, anti-bacterial, anti-cancer, and anti-protozoal activity [17,27,28]. Hence, enhydrin and uvedalin can be used as controls to determine the quality of yacon leaf extracts as a traditional medicine, in addition, the extraction technique should be considered to obtain the optimal content of the compounds.

The different ethanol solvent levels used in the extraction process affected the quantities of enhydrin and uvedalin in the yacon leaf extract. The test results showed that the higher the level of ethanol as a solvent, the higher the concentration of enhydrin compounds generated. However, it showed a different effect in uvedalin concentration, where uvedalin obtained the highest concentration in extracts with 70% ethanol as the solvent. This is because the solubility properties of enhydrin and uvedalin compounds differ. Although the structures of these compounds is very similar, the difference is in the epoxy group, where enhydrin has two epoxy groups, while uvedalin has only one. This difference is what causes the solubility level to be different. The difference in epoxy groups also affects their biological activity, which requires further research.

Therefore, the presence of these two compounds is essential as a quality control parameter for yacon leaves as medicinal plants in traditional medicine.

## 3. Materials and Methods

### 3.1. Plant Material

We collected fresh yacon leaf samplings from two different places on the island of Java, Indonesia. Samplings were taken from the Wonosobo Area, Central Java, as the first sample (Ycin). They grow on the slants of Mount Sindoro at an altitude of approximately 900 m above sea level (MASL). Other samples were taken from the Kaliurang area, Yogyakarta (Ykal), which were planted at an altitude of about 600 MASL. The Ycin sample was harvested around February–March 2021, and Ykal was collected around December 2021. The plant parts collected were old leaves with the criteria that the leaves were intact, green, and not rotten or withered. The plant age, during collection, was 3–4 months, with an average plant height of 1–1.5 m.

### 3.2. Solvent and Instrument

The solvents used for the isolation of yacon leaves were chloroform, methanol, and diethyl ethers(Merck, New York, NY, USA), all of which were standard pro-analysis. Furthermore, the analysis of HPLC used acetonitrile, methanol, and HPLC-grade water. All solvents were filtered before use. HPLC for isolation was performed using a Waters e2695(Waters corp, Wilford, MA, USA), a 2489 UV–Vis detector(Waters corp, Wilford, MA, USA), Empower 3 Software, and a Sunfire C18 column (250 × 4.6 mm; 5 μm)(Waters corp, Wilford, MA, USA).

### 3.3. Extraction and Isolation

The process of extraction and isolation of marker compounds use the method that has been carried out by previous research [15] with several modifications. It rinsed 100 g of the dried yacon leaves into the chloroform solutions for 1–2 min. The rinsed solution was filtered and then evaporated using a rotary evaporator until a thick extract was obtained. Then, it was dissolved with 35 mL of methanol and 15 mL of distilled water was slowly added. The precipitate formed was separated and then the filtrate was evaporated. The thick extract was dissolved with methanol p.a. and then stored in the freezer at −20 °C for three days to produce crystals, and then washed with cold diethyl ether three times. The crystals obtained were separated by preparative HPLC to obtain compounds **1** (56 mg) and **2** (35 mg). The solvent used for separation by preparative HPLC was acetonitrile: water in a gradient on a C18 column (150 × 7.8 mm, 5 µm).

### 3.4. Validation of HPLC Method

#### 3.4.1. Preparation of Standard Solutions (Compounds **1** and **2**)

Each standard isolate weighed as much as 10 mg and was dissolved in 10 mL of methanol to obtain a concentration of 1000 ppm. Furthermore, this standard stock solution was used to prepare a concentration series in determining the calibration curve.

#### 3.4.2. Chromatographic Conditions

The separation was performed on a Sunfire C18 column (250 × 4.6 mm, 5 μm). The mobile phase used was water (%A) and acetonitrile (%B), and gradient elution at a composition of 60% water and 40% acetonitrile was performed for up to 30 min. The injection volume was 20 µL, with a mobile phase flow rate of 1 mL/min. A UV–Vis detector was used at a wavelength of 210 nm.

#### 3.4.3. Validation Parameters

The validation method was carried out following ICH (2022) guidelines which included linearity, precision, accuracy, LOD, and LOQ. The linearity test was carried out by making a calibration curve of five series of standard concentrations (isolate 1 and 2) in the range of 5–100 ppm. A calibration curve was made by relating the concentration (*x*-axis) and peak area (*y*-axis) as the response of the HPLC chromatograms. Precision was conducted by injecting with different standard concentrations (10, 20, and 50 ppm) nine time on the same day (repeatability). Precision was expressed by the %RSD of the peak area. Accuracy was conducted by adding a standard solution, with a known concentration to the sample (spike). The concentration of the standard solution added was 5, 10, and 25 ppm in the 0.1% sample extract used. Furthermore, LOD and LOQ were determined based on the standard deviation of the response to the slope of the standard calibration curve. The LOD value was 3 times the response standard deviation (slope), while the LOQ value was ten times the response standard deviation (slope).

### 3.5. Preparation of Sample Analysis

Fresh yacon leaves (Ycin and Ykal) were cut into pieces and then dried in an oven at 50 °C for 18 h. The dried leaves were milled and sieved to obtain a powder size of 40 mesh. The extraction process for the Ycin and Ykal samples was carried out using the same technique: maceration using 50, 70, and 90% ethanol solvents by heating to 60 °C for one hour. The extract was filtered and evaporated to obtain a thick extract. The Sample were prepared by weighing the extracts and then dissolving them in methanol until a concentration of 0.1% was obtained. Ycin and Ykal samples were analyzed using the validated HPLC method to determine the extracts’ concentrations of the enhydrin and uvedalin compounds.

## 4. Conclusions

The method was found to be linear, precise, accurate, and suitable to be applied for control quality analyses of yacon leaf extracts. Therefore, we suggest that this method used for the routine analysis of enhydrin and uvedalin in yacon leaf extracts and formulations containing yacon leaf extracts.

## Figures and Tables

**Figure 1 molecules-28-01913-f001:**
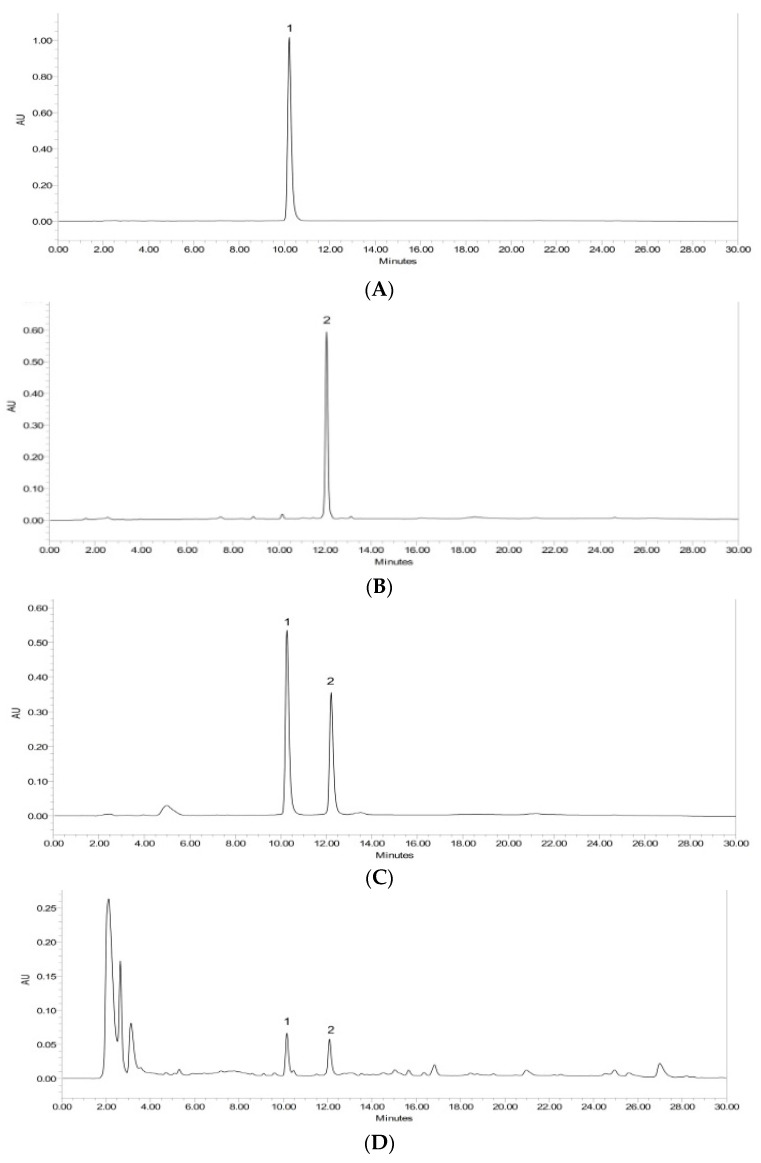
HPLC chromatogram of compound **1** (**A**), compound **2** (**B**), compound **1** and **2** combined (**C**), and ethanol extract of yacon (**D**).

**Figure 2 molecules-28-01913-f002:**
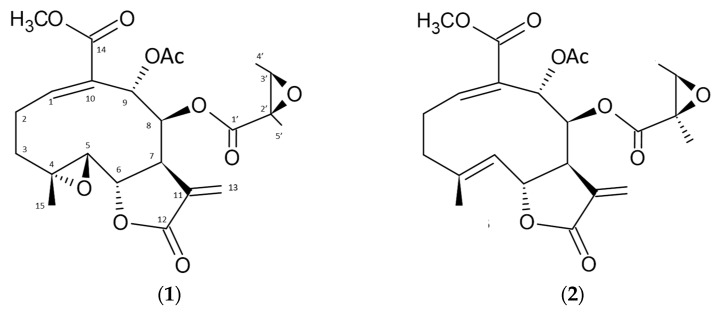
Chemical structure of enhydrin (**1**) and uvedalin (**2**).

**Table 1 molecules-28-01913-t001:** ^13^C NMR spectral data of compounds **1** and **2** compared to a standard reference [22,23].

Position	ẟC of Enhydrin		ẟC of Uvedalin	
	Compound 1	Standard Reference of Enhydrin [23]	Compound 2	Standard Reference of Uvedalin [22]
1	149.4	149.4	148.38	148.1
2	24.7	24.7	26.16	26.1
3	35.4	35.4	36.94	36.8
4	59.3	59.3	130.64	131.1
5	62.8	62.8	126.14	126.9
6	75.97	76.0	75.21	74.9
7	59.9	59.7	50.94	51.1
8	71.2	71.2	71.11	71.3
9	70.4	70.5	71.03	71.1
10	130.1	130.4	134.48	135.5
11	133.3	133.1	138.68	137.9
12	168.0	167.9	168.5	168.6
13	122.9	123.3	121.55	120.6
14	165.5	165.5	165.9	165.7
15	17.5	17.5	16.92	16.5
1′	168.4	168.6	169.1	168.7
2′	59.4	59.4	59.39	59.2
3′	45.5	45.5	59.93	59.6
4′	13.6	13.8	13.66	13.7
5′	19.1	19.2	19.14	19.3
Ac	170.4 20.8	170.1 20.9	170.2 20.93	170.0 20.3
OCH3	52.5	52.6	52.35	51.8

**Table 2 molecules-28-01913-t002:** Linear regression analysis parameters for the determination of compounds **1** and **2**.

	Concentration Range (µg/mL)	Linear Regression Parameters			LOD (µg/mL)	LOQ (µg/mL)
		Slope	y-Intercept	R^2^		
Enhydrin (**1**)	5–100	40,053	6383.6	0.9999	0.519	1.574
Uvedalin (**2**)	5–100	41,291.33	4172.92	0.9999	0.144	0.436

**Table 3 molecules-28-01913-t003:** Data of the HPLC method’s precision at repeatability levels for the compounds **1** and **2** quantifications (n = 9).

Concentration Level of the Linear Range	Amount Added (µg/mL)	Amount Found ±SD (µg/mL)	%RSD	%R
Enhydrin (**1**)	10	9.88 ± 0.33	0.34	98.82
	20	20.03 ± 0.01	0.02	100.14
	50	50.44 ± 0.27	0.53	100.88
Uvedalin (**2**)	10	10.13 ± 0.01	0.07	101.33
	20	21.11 ± 0.03	0.14	105.57
	50	51.35 ± 0.01	0.02	102.67

**Table 4 molecules-28-01913-t004:** Data on the HPLC method’s accuracy for the compounds **1** and **2** quantifications.

	Concentration Level of the Linear Range (µg/mL)	Standard Area in the Sample (AUC)	Standard Concentration in the Sample (µg/mL)	Standard Concentration (µg/mL)	%R
Enhydrin (**1**)	5	235,238	5.71	5.01	100.06
	10	442,869	10.89	10.19	101.87
	25	1,060,821	26.32	25.61	102.46
Average recovery (%RSD)					101.46 (1.23)
Uvedalin (**2**)	5	343,638	8.42	4.81	96.14
	10	554,663	13.53	9.92	99.18
	25	1,154,012	28.05	24.43	97.73
Average recovery (%RSD)					97.68 (1.55)

**Table 5 molecules-28-01913-t005:** Assay of enhydrin and uvedalin in yacon leaf ethanolic extract.

Sample	Amount of Ethanolic Extract (%*w*/*w*)	
	Enhydrin (1)	Uvedalin (2)
Ykal, extracted using 90% ethanol	1.67	0.88
Ykal, extracted using 70% ethanol	0.98	0.94
Ykal, extracted using 50% ethanol	0.29	0.39
Ycin, extracted using 90%ethanol	1.26	0.59
Ycin, extracted using 70% ethanol	0.7	0.7
Ycin, extracted using 50% ethanol	0.14	0.24

## Data Availability

The data presented in this study are available upon request from the corresponding author.

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
