# Peer review of "Quantification of Enhydrin and Uvedalin in the Ethanolic Extract of Smallanthus sonchifolius Leaves Validated by the HPLC Method"

_molecules, 2023, doi:10.3390/molecules28041913_

Round 1
Reviewer 1 Report
Quantification of Enhydrin and Uvedalin in Ethanolic Extract Of Smallanthus sonchifolius Leaves by Validated HPLC Method
Hady Anshory Tamhid, Triana Hertiani, Yosi Bayu Murti, Retno Murwanti
The manuscript describes the method for separating and determining the concentration of two sesquiterpenes with a closed structure present in Yacon leaves (Smallanthus sonchifolius).
The choice of the methodology is with HPLC, therefore the Authors proposed a method and validated it.
The manuscript is not of general interest but could be useful for Reader interested in enhydrin and uvedalin belonging to the germacrene type sesquiterpene.
The first part of the study is aimed at the isolation of these two molecules and their chemical characterization, the second part at the chromatographic methods for their quantification. The lack of data from the chromatographic part which provides the readers with only the final equations and some statistical evaluations, will more usefully have the entire dataset in the Supplementary section.
In table 2 is reported that for uvedalin the LOD is 8.423 ug/ml, but in table 4 is reported an experiment at 5ug/ml, how this datum is reliable?
Another point is that the chromatograms are missing, only two LC-MS, but no idea what happens with the UV detector. In addition are completely absent the chromatograms showing the two standard molecules together and the full chromatogram of the extract. The validation of the HPLC for the combined two molecules is not reported.
Without all this information it will be difficult to judge this manuscript, my suggestion is a thorough and major review.
Please also check for some typo mistakes. See line 192: 20 L; Table 4; and so on.
Reviewer 2 Report
The manuscript described a HPLC method for simultaneous determination of enhydrin and uvedalin in Smallanthus sonchifolius Leaves, which is helpful to control its quality. The other comments are present below:
1. The English should be improved.
2. Table 2, please check the significant figures, and add the unit of LOD, LOQ.
3. The precision and accuracy (Table 3 and 4) should be revised according the previous similar HPLC reference in Molecules. The intra-day and inter-day precision often tested by repeat injection of mixed standards in one day and three days. The repeatability usually evaluated by analysis sample 6 times. The table of recovery should present the data of found amount and added amount. In addition, the stability of sample solution should be added.
4. The chromatogram of reference compounds and sample should be added.
5. Please add more test sample, two samples is too less.
6. Line 191, “30th min”, line 192, “20L”, please checked.
7. The format of references should follow the guide of Molecules.
Round 2
Reviewer 1 Report
Quantification of Enhydrin and Uvedalin in Ethanolic Extract Of Smallanthus sonchifolius Leaves by Validated HPLC Method
Hady Anshory Tamhid, Triana Hertiani, Yosi Bayu Murti, Retno Murwanti
The amended manuscript is suitable for publication after some typo misprints to be amended.
Smallanthus sonchifolius and the other species named must be italicized throughout the manuscript.
The graphic quality of Figure 1 is very poor, it seems a cut and paste from another publication, should be improved.
Line 165 According to Wikipedia Mount Cindoro (Central Java) should be Mount Sindoro
Author Response
Response to Reviewer 1 Comments
Point 1 : Smallanthus sonchifolius and the other species named must be italicized throughout the manuscript.
Respon 1 : We have checked and italicized for some species named on the manuscript
Point 2 : The graphic quality of Figure 1 is very poor, it seems a cut and paste from another publication, should be improved.
Respon 2: Thank you for pointing this out. We have revised the graphic quality of figure 1
Point 3 : Line 165 According to Wikipedia Mount Cindoro (Central Java) should be Mount Sindoro
Respon 3: Thank you for this suggestion. We have checked and revised the name of Mount Sindoro
Reviewer 2 Report
1. Please check the linear range (5-100) and LOQ (8.423). In generally, LOQ should lower than the low concentration of linear range.
2. Please add the RSD of accuracy (Table 4)
3. Please add the stability of sample solution.
4. Line 191, “30th min”, line 192, “20L”, please check them.
Author Response
Response to Reviewer 2 Comments
Point 1 : Please check the linear range (5-100) and LOQ (8.423). In generally, LOQ should lower than the low concentration of linear range.
Respon 1: Thank you for this suggestion. The reviewer is correct, and we have checked and corrected it carefully, and it turned out that there was an error in our formula writing in excel. Then we reformulated to calculate the LOD and LOQ values according to the correct approach. We obtained a LOD and LOQ value was 0.144 and 0.436, respectively. We have revised it on the manuscript. These values are lower than the lowest concentration of the linear range we used (5-100).
Point 2 : Please add the RSD of accuracy (Table 4)
Respon 2: We have added the %RSD of accuracy to the manuscript on the table 4
Pont 3 : Please add the stability of sample solution.
Respon 3: Thank you for pointing this out. Although we agree that this is an important consideration, it is not essential for inclusion in this manuscript because we only did a simple preparation, dissolved the sample with solvent ( in a particular concentration), filtered, and directly injected it in the HPLC device. Thus there is no incubation period or delay time in the process. We, therefore, do not analyze the stability of the sample.
Pont 4 : Line 191, “30th min”, line 192, “20L”, please check them.
Respon 4: We have checked and corrected for some typo mistakes